Corrected: Publisher correction

# Novel pleiotropic risk loci for melanoma and nevus density implicate multiple biological pathways

David L. Duffy [1], Gu Zhu[1], Xin Li[2], Marianna Sanna[3], Mark M. Iles[4], Leonie C. Jacobs[5], David M. Evans[6,7], Seyhan Yazar [8], Jonathan Beesley[1], Matthew H. Law [1], Peter Kraft[9], Alessia Visconti [3], John C. Taylor [4], Fan Liu[10], Margaret J. Wright [1], Anjali K. Henders[1,17], Lisa Bowdler[1], Dan Glass[3], M. Arfan Ikram [11], André G. Uitterlinden[11,12], Pamela A. Madden[13], Andrew C. Heath[13], Elliot C. Nelson[13], Adele C. Green [1,14], Stephen Chanock [15], Jennifer H. Barrett [4], Matthew A. Brown [7], Nicholas K. Hayward[1], Stuart MacGregor [1], Richard A. Sturm[16], Alex W. Hewitt [8], Melanoma GWAS Consortium[#], Manfred Kayser [10], David J. Hunter[9], Julia A. Newton Bishop[4], Timothy D. Spector[3], Grant W. Montgomery [1,17], David A. Mackey [8], George Davey Smith[6], Tamar E. Nijsten[5], D. Timothy Bishop [4], Veronique Bataille[3], Mario Falchi[3], Jiali Han[2] & Nicholas G. Martin[1]

The total number of acquired melanocytic nevi on the skin is strongly correlated with melanoma risk. Here we report a meta-analysis of 11 nevus GWAS from Australia, Netherlands, UK, and USA comprising 52,506 individuals. We confirm known loci including *MTAP*, *PLA2G6*, and *IRF4*, and detect novel SNPs in *KITLG* and a region of 9q32. In a bivariate analysis combining the nevus results with a recent melanoma GWAS meta-analysis (12,874 cases, 23,203 controls), SNPs near *GPRC5A*, *CYP1B1*, *PPARGC1B*, *HDAC4*, *FAM208B*, *DOCK8*, and *SYNE2* reached global significance, and other loci, including *MIR146A* and *OBFC1*, reached a suggestive level. Overall, we conclude that most nevus genes affect melanoma risk (*KITLG* an exception), while many melanoma risk loci do not alter nevus count. For example, variants in *TERC* and *OBFC1* affect both traits, but other telomere length maintenance genes seem to affect melanoma risk only. Our findings implicate multiple pathways in nevogenesis.

[1] QIMR Berghofer Medical Research Institute, Brisbane, Australia. [2] Department of Epidemiology, Richard M. Fairbanks School of Public Health, Melvin and Bren Simon Cancer Center, Indiana University, Indianapolis, IN 63110, USA. [3] Department of Twin Research & Genetic Epidemiology, St Thomas Hospital Campus, Kings College, London, UK. [4] Section of Epidemiology and Biostatistics, Leeds Institute of Cancer and Pathology, University of Leeds, Leeds, UK. [5] Department of Dermatology, Erasmus MC University Medical Centre Rotterdam, Rotterdam, The Netherlands. [6] MRC Integrative Epidemiology Unit, University of Bristol, Bristol, UK. [7] University of Queensland Diamantina Institute, Translational Research Institute, Brisbane, Australia. [8] Centre for Ophthalmology and Vision Science, University of Western Australia and the Lions Eye Institute, Perth, Australia. [9] Department of Epidemiology, Harvard T.H. Chan School of Public Health, Boston 02115 MA, USA. [10] Department of Genetic Identification, Erasmus MC University Medical Centre Rotterdam, Rotterdam, The Netherlands. [11] Department of Epidemiology, Erasmus MC, Rotterdam, Netherlands. [12] Department of Internal Medicine, Erasmus MC, Rotterdam, Netherlands. [13] Department of Psychiatry, Washington University School of Medicine, St. Louis, MO 63110, USA. [14] Molecular Oncology Group, CRUK Manchester Institute, University of Manchester, Manchester, UK. [15] Division of Cancer Epidemiology and Genetics, National Cancer Institute, Bethesda, MD, USA. [16] Dermatology Research Centre, University of Queensland Diamantina Institute, Translational Research Institute, Brisbane, Australia. [17] Present address: Institute for Molecular Bioscience, The University of Queensland, Brisbane, Australia. These authors contributed equally: Jiali Han and Nicholas G. Martin. Full list of members of the Melanoma GWAS Consortium is given at the end of this paper. Correspondence and requests for materials should be addressed to D.L.D. (email: David.Duffy@qimrberghofer.edu.au)

The incidence of cutaneous malignant melanoma (CM) has increased in populations of European descent in North America, Europe, and Australia due to long-term changes in sun exposure behavior, as well as screening[1]. The strongest CM epidemiological risk factor acting within populations of European descent is the number of cutaneous acquired melanocytic nevi, with risk increasing by 2–4% per additional nevus counted[2]. Nevi are benign melanocytic tumors usually characterized by a signature somatic *BRAF* mutation. Their association with CM can be direct, in that a proportion of melanomas arise within a pre-existing nevus (due to a "second hit" mutation), or indirect, where genetic or environmental risk factors for both traits are shared. Total nevus count is highly heritable (60%–90% in twins)[3,4], but only a small proportion of this genetic variance is explained by loci identified so far[5–9]. The known nevus count loci all have pleiotropic effects on CM risk[5–9], which implies both that nevus count loci are medically important and that a genetic analysis combining nevi and CM phenotypes will have increased statistical power. Here we present a new large nevus genome-wide association meta-analysis, and combine these results with those of a previously published meta-analysis of melanoma[10].

## Results

**Nevus GWAS meta-analysis.** Genome-wide single-nucleotide polymorphism (SNP) genotype data were available for a total of 52,806 individuals from 11 studies in Australia, UK, USA, and the Netherlands (Table 1), where nevus number had been measured by counting or ratings, by self or observer, and of the whole body or selected regions. Analyses show that these are measuring the same entity and are therefore combinable for GWAS (genome-wide association study; see Supplementary Results). The genomic inflation factors were $\lambda = 1.41$ and $\lambda_{1000} = 1.008$ (Q–Q plot, Supplementary Fig. 1), consistent with polygenic inheritance and the total sample size[11]. Five genomic regions contained association peaks that reached genome-wide significance in the nevus count meta-analysis (Fig. 1, Table 2, Supplementary Fig. 2),

*MTAP/CDKN2A* on chromosomes 9p21.3 (peak SNP, $P = 2 \times 10^{-37}$) and 9q31.1-2 ($P = 1 \times 10^{-8}$), *IRF4* on chromosome 6p (peak SNP, $P = 4 \times 10^{-37}$), in *KITLG* in the region of the known testicular germ cell cancer risk locus ($P = 8 \times 10^{-9}$), rs600951 over *DOCK8* on chromosome 9p24.3 ($P = 2 \times 10^{-8}$), and *PLA2G6* on chromosome 22 ($P = 3 \times 10^{-18}$). We have previously detected three of these in analyses using subsets of the meta-analysis sample[5,10]. A SNP, rs251464, in *PPARGC1B* ($P = 5 \times 10^{-7}$), reached a suggestive level of association. We detected statistical heterogeneity in association with nevus count especially for *IRF4*, *MTAP*, *PLA2G6*, and *DOCK8* (see Supplementary Tables 1 and 2) —that for *IRF4* was expected—given our original studies of this gene showing crossover G × age interaction.[10] Meta-regression including age of the current study participants confirmed the age effect in the case of *IRF4* (Supplementary Table 1).

**Combining nevus and melanoma GWAS meta-analyses— Bayesian analysis.** We then combined these nevus meta-analysis *P* values with those from the melanoma meta-analysis[10] (Table 1, Fig. 2, Supplementary Figs 1, 2). We used simple combination of *P* values (weighted Stouffer method), as well as the GWAS-PW program,[12] which combines GWAS data for two related traits to investigate the causes of genetic covariation between them (see Supplementary Methods). Specifically, it estimates Bayes factors and posterior probabilities of association (PPA) for four hypotheses: (a) a locus specifically affects melanoma only or (b) affects nevus count only; (c) a locus has pleiotropic effects on both traits; and (d) there are separate alleles at a locus independently determining each trait (colocation).

There were 30 regions containing SNPs that met our threshold for "interesting" (PPA > 0.5) for any of these hypotheses (Fig. 3, Supplementary Table 3). Twelve of these loci exhibited no evidence of association to nevus count, but were strongly associated with melanoma risk, one of the most extreme being *MC1R*. A total of 18 loci showed pleiotropic action with consistent directional and proportional effects of all SNPs on

**Table 1 GWAS studies of nevus count contributing to the present meta-analysis**

| Study | Nevus assessment | SNP chip | Imputation | Individuals (families) | Age range (mean) | Location (center) |
|---|---|---|---|---|---|---|
| ALSPAC[39] | Self-count on limbs | 550k | 1000Gv.3 | 3309 | 14–17 (15.5) | UK (Bristol) |
| Harvard[8] | Self-count >3 mm on limbs | Affy+Illumina various | 1000Gv.3 | 32,975 | 35–75 (52) | US (Boston) |
| Leeds[40] | Whole-body count >2 mm | OmniExpressExome | HRC v.1 | 397 | 21–80 (57) | Yorkshire |
| QIMR BTNS children[3] | Whole-body count >0 mm | 610k, CoreExome | 1000Gv.3 | 3261 (1309) | 9–23 (12.6) | SE Queensland (Brisbane) |
| QIMR BTNS parents[9] | Self-rating 4-point scale | 610k+CoreExome | 1000Gv.3 | 2248 (1299) | 29–72 (44.1) | SE Queensland |
| QIMR adult twins[41] | Self-rating 4-point scale | 317k+370k+610k +CE | 1000Gv.3 | 1848 (1113) | 29--79 (52.3) | Australia wide |
| QIMR >50 twins[42] | Self-count right arm >4 mm | 370k+610k+CE | 1000Gv.3 | 893 (596) | 50–92 (60.7) | Australia wide |
| Raine[43] | Nurse-count right arm | 660k | 1000Gv.3 | 808 | 22 | Western Australia (Perth) |
| Rotterdam[44] | Whole-body rating 4-pt scale | 550k, 610k | 1000Gv.3 | 3319 | 51–98 (67) | Rotterdam (NL) |
| TEST[45] | Whole-body count >0 mm | 610k+CE | 1000Gv.3 | 136 (71) | 5–18 (9.7) | Tasmania+Victoria |
| Twins UK[5] | Whole-body count >2 mm | 317k+610k+1M +1.2M | 1000Gv.3 | 3312 (1839) | 18–80 (47) | UK wide (London) |
| Total nevus | | | | 52,506 | | |
| Melanoma GWASMA[10] | | | | 12,874 cases; 23,203 controls | | |
| Nevus+melanoma | | | | 88,583 (inc. controls) | | |

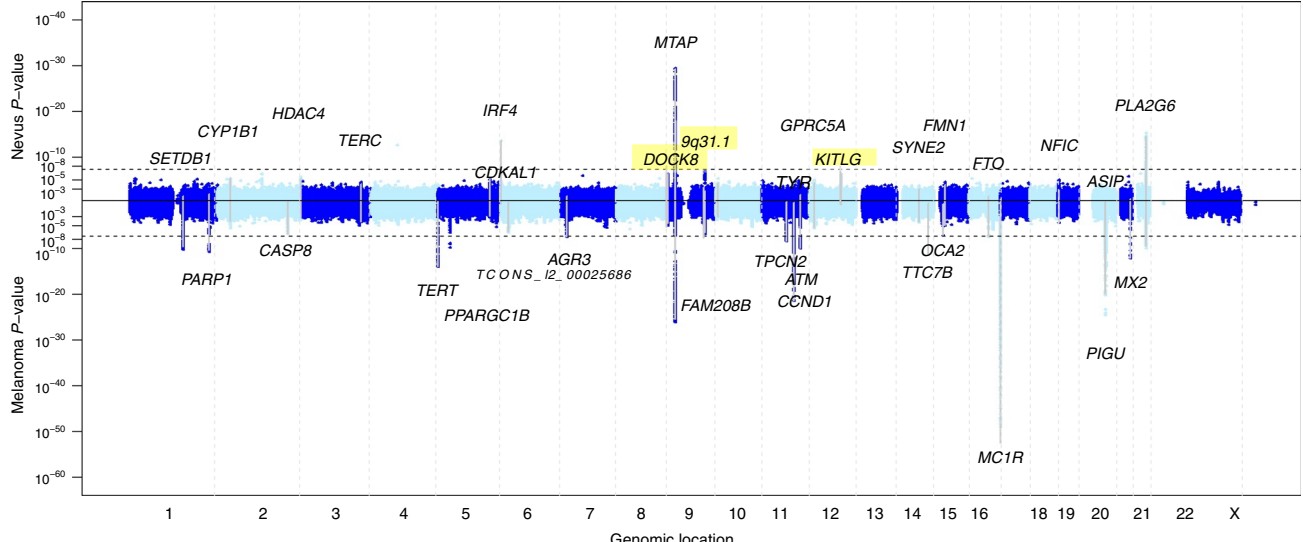

**Fig. 1** Miami plot of nevus count and melanoma meta-analysis. $P$ values where either $P < 10^{-5}$. The –log10 $P$ values for the nevus GWAS meta-analysis are above the central solid line and those for the melanoma GWAS meta-analysis are below that line. Novel nevus loci are highlighted

| SNP | Position (hg19) | Combined P | CM P | Nevus P | Gene/interval |
|---|---|---|---|---|---|
| rs869329 | 9:21804693 | 7.48E−67 | 1.14E−31 | 2.12E−37* | MTAP |
| rs11532907 | 9:21844772 | 1.72E−34 | 1.42E−19 | 2.30E−17 | |
| rs132985 | 22:38563471 | 2.07E−28 | 4.76E−12 | 3.06E−18* | PLA2G6 |
| rs2005974 | 22:38537112 | 3.30E−23 | 7.83E−11 | 3.31E−14 | |
| rs12203592 | 6:396321 | 5.84E−01 | 8.22E−01 | 4.21E−67* | IRF4 |
| rs7313352 | 12:88949124 | 2.27E−05 | 6.61E−01 | 8.40E−09 | KITLG** |
| rs600951 | 9:224742 | 9.89E−13 | 5.52E−06 | 1.95E−08* | DOCK8** |
| rs10816595 | 9:110709735 | 1.70E−14 | 1.49E−07 | 1.08E−08 | 9q31.2 |
| rs251464 | 5:149196234 | 1.92E−09 | 4.58E−04 | 4.71E−07 | PPARGC1B |
| rs4670813 | 2:38317710 | 1.14E−10 | 2.40E−05 | 5.70E−07 | CYP1B1 |
| rs1640875 | 12:13069524 | 3.30E−11 | 4.08E−07 | 5.72E−06* | GPRC5A** |
| rs1148732 | 12:13068291 | 1.08E−09 | 2.08E−04 | 6.21E−07 | |
| rs55875066 | 2:240076002 | 1.35E−09 | 2.16E−04 | 7.59E−07 | HDAC4** |
| rs12696304 | 3:169481271 | 8.30E−10 | 1.64E−05 | 5.73E−06 | TERC |
| rs117648907 | 15:33277710 | 1.13E−10 | 1.43E−06 | 6.52E−06 | FMN1** |
| rs45575338 | 10:5784151 | 2.16E−08 | 2.87E−04 | 1.02E−05 | FAM208B** |
| rs1484375 | 9:109067561 | 1.56E−10 | 2.30E−08 | 1.35E−04 | 9q31.1 |
| rs2357176 | 14:64409313 | 3.89E−08 | 1.74E−05 | 1.95E−04 | SYNE2** |
| rs34466956 | 19:3353622 | 2.92E−08 | 1.02E−05 | 2.22E−04 | NFIC** |
| rs1636744 | 7:16984280 | 1.29E−09 | 1.84E−09 | 0.002 | TCONS_l2_00025686 |
| rs380286 | 5:1320247 | 3.18E−14 | 1.66E−17 | 0.003* | TERT |
| rs2695237 | 1:226603635 | 1.49E−11 | 3.59E−13 | 0.004 | PARP1 |
| rs73008229 | 11:108187689 | 8.21E−11 | 1.38E−12 | 0.006 | ATM |
| rs72704658 | 1:150833010 | 1.90E−10 | 3.88E−12 | 0.007 | SETDB1 |
| rs12596638 | 16:54115829 | 2.30E−08 | 1.81E−09 | 0.014 | FTO |
| rs416981 | 21:42745414 | 3.90E−10 | 3.28E−15 | 0.063 | MX2 |
| rs75570604 | 16:89846677 | 1.64E−45 | 6.24E−92 | 0.067 | MC1R |
| rs7582362 | 2:202176294 | 4.32E−06 | 8.88E−09 | 0.134 | CASP8 |
| rs498136 | 11:69367118 | 1.42E−06 | 1.01E−10 | 0.209 | TPCN2/CCND1 |
| rs56238684 | 20:33236696 | 5.14E−13 | 8.36E−25 | 0.215 | ASIP |
| rs2125570 | 6:21166705 | 9.14E−05 | 3.27E−08 | 0.351 | CDKAL1 |
| rs184628474 | 14:91185865 | 4.32E−07 | 4.63E−14 | 0.415 | TTC7B |
| rs10830253 | 11:89028043 | 2.32E−11 | 1.01E−26 | 0.605 | TYR |
| rs250417 | 5:33952378 | 5.18E−05 | 2.30E−12 | 0.755 | SLC45A2 |
| rs4778138 | 15:28335820 | 5.52E−03 | 3.11E−09 | 0.935 | OCA2 |

**Table 2 SNPs associated with total nevus count and cutaneous melanoma (CM) in their respective meta-analyses**

The weighted Stouffer method was used to combine the nevus and melanoma $P$ values (Combined $P$). The SNP with the smallest combined $P$ value under each peak is shown, but the table rows are ordered by strength of association to nevus count. In three cases where significant between-study heterogeneity is detected (unadjusted $P_{hom} < 0.05$, denoted by *), the nevus $P$ value is from the random-effects model of Han and Eskin[38], and a result for a nearby SNP where $P_{hom} > 0.05$ is included on the line beneath (*italicized*) to confirm genome-wide significance (in the case of *IRF4* and *DOCK8*, there is no such nearby SNP).
*Unadjusted $P_{hom} < 0.05$
**Novel loci

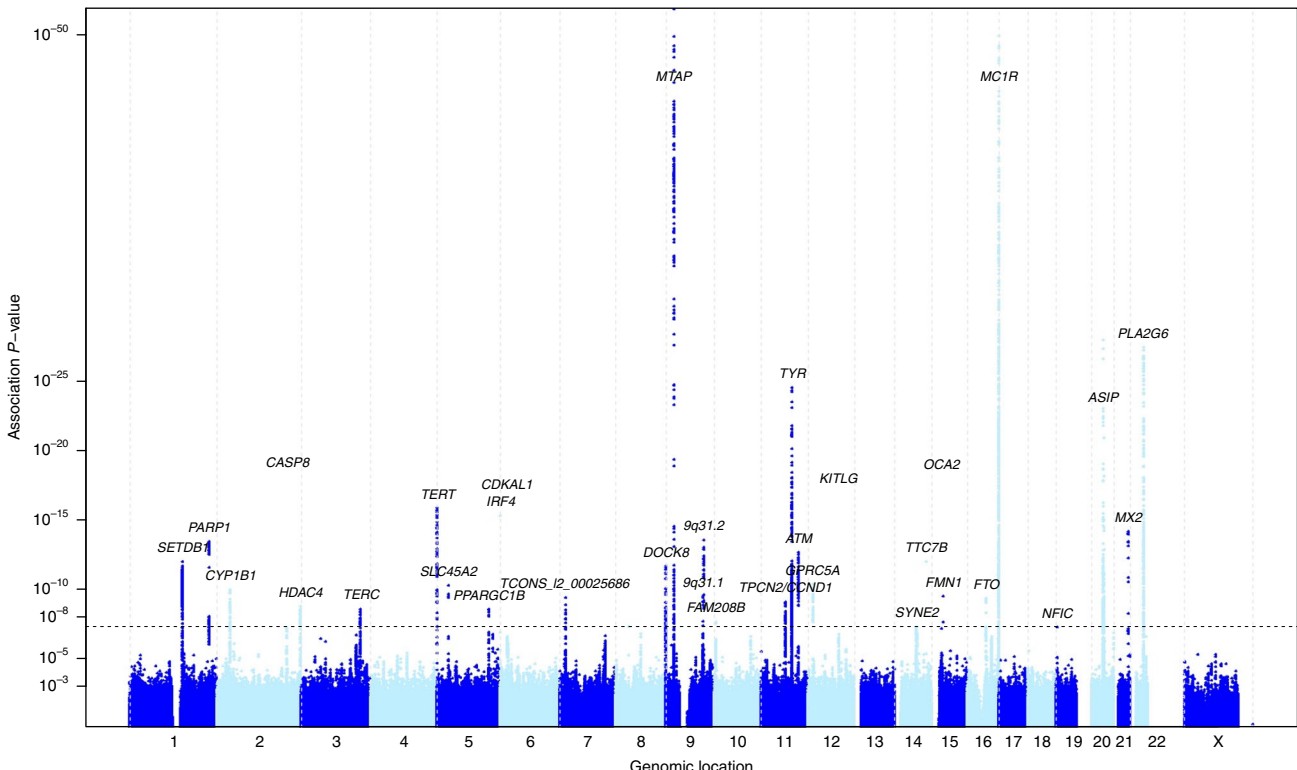

**Fig. 2** Manhattan plot of *P* values from meta-analysis combining nevus and melanoma results

nevi and melanoma risk, the strongest being *MTAP, PLA2G6*, and an intergenic region on 9q31.1 (Fig. 4a shows a bivariate regional association around *GPRC5A*, all loci are shown in Supplementary Figs 5–19). There were no "pure nevus" regions using the binned GWAS-PW test (hypothesis b, PPAb > 0.2), with even the region of *KITLG* appearing as a pleiotropic region (PPAb = 0.52, PPAc = 0.11), even though the pattern of bivariate association appears more consistent with a "nevus-only" locus (Fig. 4b). For another five regions, support was split between the pure melanoma and pleotropic models. In the case of *IRF4*, this is certainly driven by the marked between-study heterogeneity in melanoma association due to their different age distributions and latitudinal origins[13].

One interesting SNP (rs34466956), 2 kbp upstream from *NFIC* on chromosome 19p13.3 (see Fig. 5), achieved a combined *P* value of $3 \times 10^{-8}$ and a SNP-wise PPAc for pleiotropism of 0.9, even though the binned GWAS-PW assigned the region a highest PPA of 0.28.

**Pleiotropy.** The 18 pleiotropic loci each come from multiple pathways, indicating that nevogenesis is a more complicated process than previously anticipated. Pathways already implicated include those of *MTAP* (purine salvage pathway, possibly a rate limiting step to cell proliferation), *PLA2G6* (phospholipase A2, implicated in apoptosis), and *IRF4* (melanocyte pigmentation and proliferation). Newly implicated here in nevogenesis, *TERC* is a strong candidate given its involvement in telomere maintenance and prior suggestive evidence of association with melanoma/ nevi[10,14,15], as well as several other cancers.[16–18] *PPARGC1B* has previously been investigated as a skin color locus[17] and there is functional evidence for its effects on melanocytes.[18] *GPRC5A* (see Fig. 4a, Supplementary Fig. 15) has also been suggestively associated with melanoma[10] and is a known oncogene in breast and lung cancer[19,20]. *DOCK8* deficiency predisposes to virus-related

malignancy and is deleted in some cancers, but not markedly in melanoma.[21,22] DOCK8 regulates Cdc42 activation especially in immune effector cells—Cdc42 has been implicated in melanoma invasiveness[23] and variants in *CDC42* have been previously associated with melanoma tumor thickness[24] —though our best association *P* value in the region of that latter gene is $3 \times 10^{-4}$.

The novel pleiotropic loci are: (a) the region around *HDAC4* on chromosome 2; (b) chromosome 9q31 (two separate peaks); (c) near *SYNE2* on chromosome 14; (d) in *DOCK8* on chromosome 9p; and (e) near *FMN1* on chromosome 15p (see Supplementary Results). For those loci that unequivocally lie within a gene, in each case that gene is expressed in melanocytes[25] and these implicate several different pathways. The "master regulator" in melanocytogenesis[26] is MITF (microphthalmia-associated transcription factor), and we confirmed that our top candidate genes in each of the 30 regions contain MITF binding sites.[27] For example, three genes in the *FMN1* region harbor MITF binding sites, viz. *SCG5, RYR3*, and *FMN1* themselves (enrichment *P* = 0.01). Furthermore, in several of these genes (*MTAP, IRF4, PLA2G6, GPRC5A*, and *TERC*), the most associated SNP lies within or close to the actual MITF binding sites, in some cases a rarer MITF–BRG1–SOX10–YY1 combined regulatory element (MARE)[27] (Supplementary Figs 20–40).

**Gene based tests.** The genes most strongly implicated in a gene-based association analysis (PASCAL) are *MTAP, PLA2G6, GPR5A, ASB13* (adjacent to *FAM208B*), and *KITLG* ($P = 2.3 \times 10^{-6}$); see Supplementary Table 4. At a suggestive level, we note *FAM208B, MGC16025* (both $P = 6 \times 10^{-6}$), and *HDAC4* ($1 \times 10^{-5}$). Among genes at a significance level of $<10^{-4}$, we highlight *LMX1B* ($P = 5 \times 10^{-5}$), where rs7854658 gave a nevus *P* value of $3.3 \times 10^{-6}$.

**Pathway analysis.** Using different approaches (GWAS PRS, GWAS-PW, and REML using SNP sets; see Supplementary

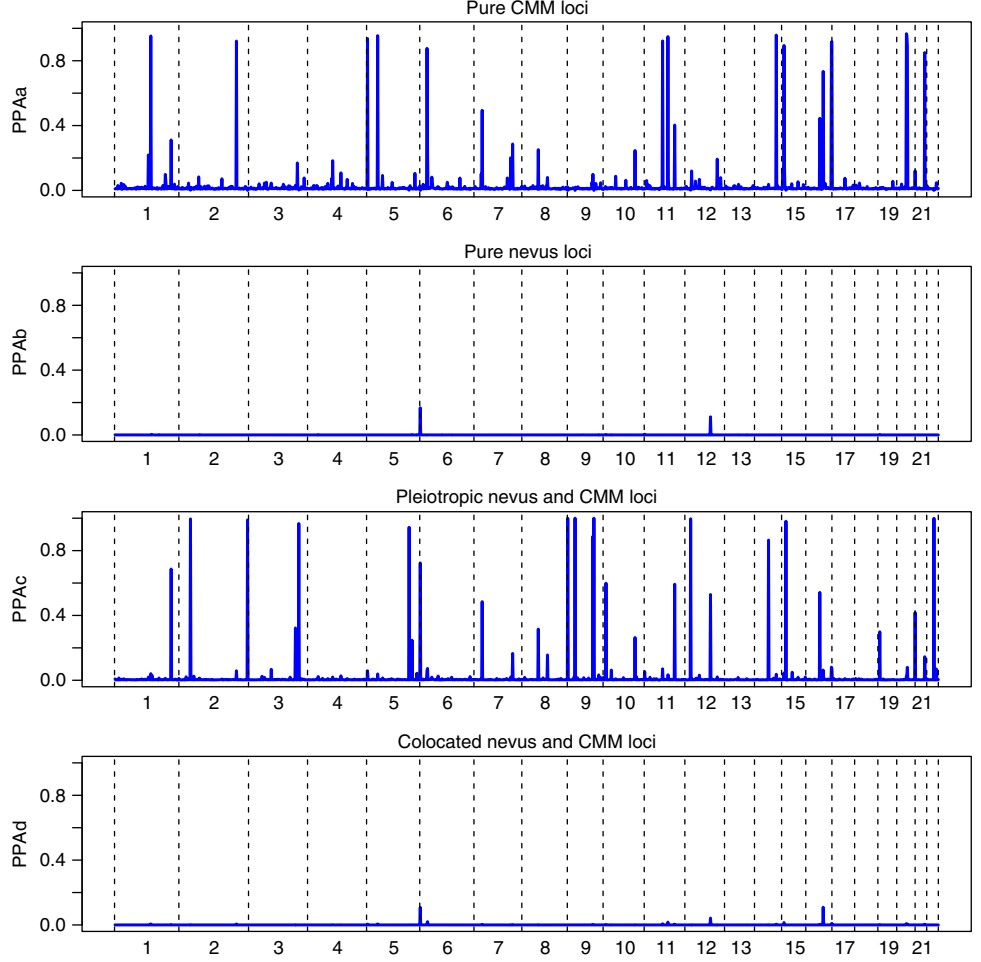

**Fig. 3** Results of analyses using GWAS-PW, which assign posterior probabilities (PPA) to each of ~1700 genomic regions that is **a** a pure melanoma locus, **b** a pure nevus locus, **c** a pleiotropic nevus and melanoma loci, and **d** that the locus contains co-located but distinct variants for nevi and melanoma

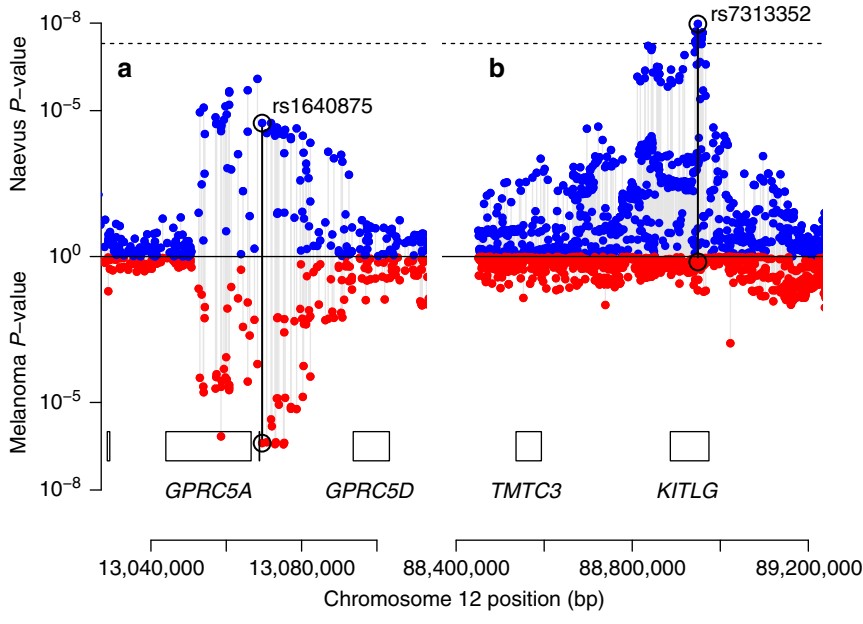

**Fig. 4** Plot of nevus and melanoma association test *P* values for **a** the region around rs1640875 in *GPRC5A* (chr12:12.9 Mbp) illustrating symmetrical influence on nevus count and melanoma risk; note that neither univariate peaks achieve significance alone but in combination they do (see Table 2, Fig. 2), and **b** the region around rs7313352 in *KITLG* (chr12:88.6 Mbp), a "pure" nevus locus with negligible direct effect on melanoma risk

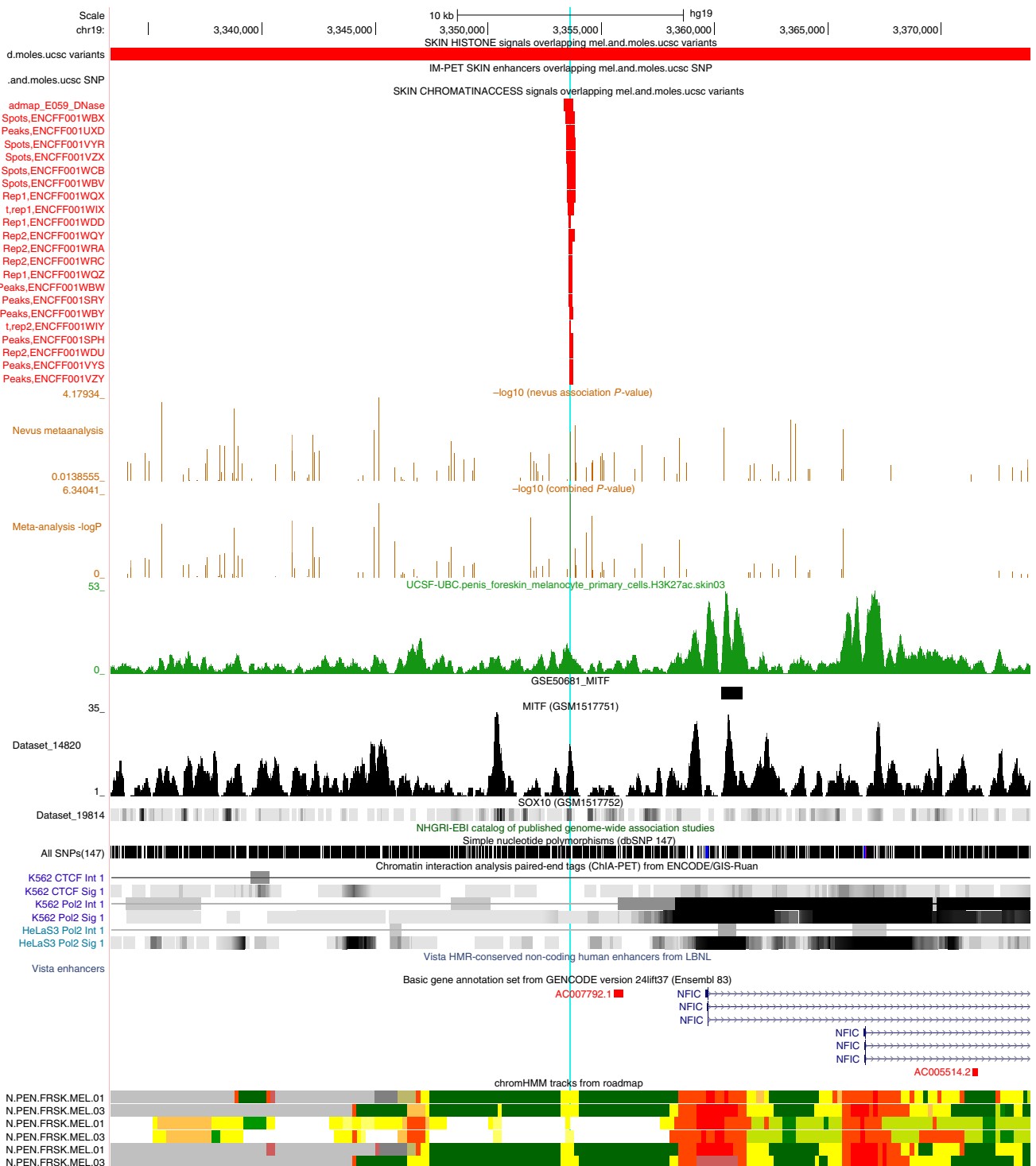

**Fig. 5** UCSC Genome Browser view of region near *NFIC* (19p13.3). The pale blue line highlights location of rs34466956, which coincides with a narrow regulatory region as seen in in the 22 short red bars indicating open chromatin in melanocytes and skin. These align in the bottom 6 tracks with narrow yellow regions indicating results of hidden Markov models summarizing the evidence from multiple experiments for open chromatin in melanocytes. An MITF ChipSeq peak also overlies this same region (gray track, GSM1517751). *NFIC* is expressed in melanocytes, and a second larger MITF peak overlies intron 1 in two ChipSeq experiments viz. GSE50681_MITF, see short solid black bar, and also the tall sharp gray peak below it in GSM1517751. See Supplementary Methods for details

Table 5), we tested candidate pathways[28] for their overall contribution to variance in nevus number, the contribution of the telomere maintenance pathway was 0.8%. A contribution of the immune regulation/checkpoint pathway was surprisingly absent, given our knowledge that immunosuppression increases nevus count quite promptly and the recent success of CTLA4 inhibitors in the treatment of melanoma. We did see a weak signal (Combined $P = 1 \times 10^{-7}$) for rs870191, very close to SLE-associated SNPs just upstream from MIR146A, an important immune regulator.

**Genetic relationships with telomere length and pigmentation.** In the GWAS-PW analysis combining melanoma and telomere length (TL) (see Supplementary Methods), there was considerable locus overlap, while by contrast only *TERC* was detectably shared between nevus count and TL (Supplementary Fig. 41). Note that SNPs in *OBFC1* were only significantly associated with melanoma in the phase 2 analysis of Law et al.[10]—which are not utilized in the GWAS-PW analysis—although they were suggestively associated ($P = 10^{-5}$) with nevus count. In the parallel analysis with pigmentation (indexed by dark hair color), only *IRF4* overlapped with nevus count (Supplementary Fig. 42). Again, multiple pigmentation loci acted as risk factors for melanoma (with no overlap with TL). The fact that only *TERC* (and *OBFC1*) are associated with nevus count, while multiple loci are associated with melanoma, is not necessarily surprising. Telomere maintenance may predispose to melanoma directly as well as via nevus count, an extension of the "divergent pathway" hypothesis for melanoma[29]. However, the link with telomere length-associated SNPs may need a bigger sample size to look at associations further.

**SNP heritability and genetic correlation.** Mixed-model twin analyses with GCTA and LDAK (see Supplementary Methods) utilizing the Australian and British samples estimate the total heritability of nevus count to be 58% (and family environment 34%), with contributions from every chromosome and one-sixth from chromosome 9 alone (see Supplementary Table 6). We found that ~25% of the Australian and ~15% of British genetic variance for nevus count could be explained by a panel of 1000 SNPs covering our 32 regions. We have also performed analyses examining the overall architecture of the relationship between nevus count and melanoma risk using bivariate LD score regression analysis and estimated $r_g = 0.69$ ($SE = 0.16$) (see Supplementary Results). Alleles which increase nevus number proportionally increase the risk of melanoma (Supplementary Results, Supplementary Figs 43, 44) with *KITLG,* the interesting exception is that the nevus-associated variants did not predict melanoma risk (see Fig. 5b), rather, predisposing to other cancers (e.g., testicular germ cell).

## Discussion

It has been long suggested that carrying out genetic analyses using multiple correlated phenotypes will increase power to detect trait loci in such a way as to justify the statistical complications. Since number of cutaneous nevus is strongly correlated with melanoma risk, and known nevus loci were associated with CM, it seemed likely that this would be a fruitful approach. We have highlighted eight novel loci, including the genes *HDAC4, SYNE2*, and most notably *GPRC5A*, where quite large samples of melanoma cases or nevus count were not sufficiently powerful to reach formal genome-wide significance in univariate analyses, but the combined evidence is conclusive.

Given that lighter skin color is also associated with both these phenotypes, we would expect a strong contribution from pigmentation pathway genes. Among those novel pleiotropic loci implicated in nevus count, *CYP1B1* and *PPARGC1B* both appear in a recent skin pigmentation meta-analysis[30] as harboring variants lightening skin color. The SNPs in the chromosome 7p21.1 region near *AHR* and *AGR3* previously associated with CM also appear to be associated with skin color in that study. In our analysis, the signal for nevus count from that interval (best $P = 3 \times 10^{-4}$) was half as strong as that for CM, and the GWAS-PW analysis support was equal for the hypotheses of a pure CM locus and a pleiotropic locus (region PPAa = 0.494, PPAc = 0.485). In passing, the peak SNPs lie within a long noncoding RNA gene

(TCONS_I2_00025688) that is expressed in melanocytes, so this is a potential candidate for both skin color and CM. In the case of *KITLG*, the variant most strongly associated with pigmentation (fair hair), rs12821256, modifies a distant enhancer, and was associated neither with melanoma or nevus count in our study (see Supplementary Results). We observe a similar pattern (association $P_{nevus} = 0.4$, $P_{CM} = 0.8$) for the strongest associated variant for skin color from the skin color meta-analysis, rs11104947.[30]

By contrast, *HDAC4* and *DOCK8* are in pathways that have not been implicated as important to nevogenesis or melanoma pathogenesis. HDAC4 is involved in transcriptional regulation in many tissues, while DOCK8 acts to regulate signal transduction, most notably in immune effector cells (see Supplementary Results). The association peak for *HDAC4* is quite wide (~80 kbp), and overlaps with the multi-tissue GTEx eQTL peak for this gene.[31] The best overlapping SNP was rs115253975, with a combined nevus-CM $P$-value of $4 \times 10^{-9}$ and fibroblast *HDAC4* eQTL $P$-value of $2 \times 10^{-5}$. The peak nevus-CMM *DOCK8* SNP, rs600951, is a cis-eQTL in two (non-cutaneous) tissues, and the peak around it contains several eQTL SNPs detected in the GTEx skin samples. These eQTL SNPs would be potential causal candidates.

Both *SYNE2* (encoding nesprin-2) and *FMN1* (formin-1) are involved in nuclear envelope and cytoskeleton function, and through this in regulating as well as facilitating numerous biological pathways. Both, for example, are involved in directed cell migration. The nesprin and formin families have been implicated in efficient repair of double strand DNA breaks, so this might point to a mechanism for an association with nevi and CM (see Supplementary Results).

We did see heterogeneity between studies in strength of SNP association with nevus count or melanoma for four loci, most extremely for *IRF4* (Supplementary Fig. 10). Meta-regression analysis suggested this is partly due to interactions with age in the case of *IRF4* (Supplementary Table 1)—different nevus subtypes are known to predominate at different ages, with the dermoscopic globular type most common before age 20.[32] We suspect sun exposure another important interacting covariate, given large differences in total nevus count by latitude.[33,34]

Epidemiologically, the etiology of melanoma has been divided[35] into a chronic sun-exposure pathway and a nevus pathway, where intermittent sun exposure is sufficient to increase risk. At a genetic level, pigmentation genes such as *MC1R* contribute only via the former pathway (though this can include effects on DNA repair[36]), others such as *MTAP* via the latter, while yet others such as *IRF4* seem to act via both routes[13]. We interpret our results as consistent with the hypothesis that nevus number is the intermediate phenotype in a causative chain to melanoma originating in all these biologically heterogeneous nevus pathways. However, we acknowledge that there may also be some genes where there is a direct causal pathway to both phenotypes.

## Methods

We carried out a meta-analysis of 11 sizeable GWAS of total nevus count in populations from Australia, Netherlands, Britain, and the United States, subsets of which have been reported on previously[5,6,8], and then combined these results with those from a recently published meta-analysis of melanoma GWAS[10] to increase power to detect pleiotropic genes. While nevus counts or density assessments are available for melanoma cases from a number of studies, in the meta-analysis of nevus count we included only samples of healthy individuals without melanoma, all of European ancestry (for more details, see Supplementary Methods).

**Nevus phenotyping.** The assessment of nevus counts varies considerably between the 11 studies in four respects (see Table 1): (a) nevus counts vs. density ratings; (b) whole body vs. only certain body parts; (c) all moles (>0 mm diameter) or only

moles >2 mm, or 3 mm, or 5 mm; and (d) count by trained observer or self-count by study participant. These differences could contribute statistical heterogeneity into our analyses, so we have done considerable preliminary work to convince ourselves that all assessments are measuring the same biological dimension of "moliness" (see Supplementary Fig. 3). A pragmatic test of this is the relative contribution of each study to the detection of the known loci of large effect, which is evident from the forest plots (Supplementary Figs 5–19).

**Statistical methods**. Given this, we combined results from each study as regression coefficients and associated standard errors in standard fixed and random effects meta-analyses using the METAL[37] and METASOFT[38] programs. Manhattan and Q–Q plots for the nevus GWAS meta-analysis (GWASMA) are shown in Supplementary Fig. 45 and for each of the contributing studies in Supplementary Figs 46–55.

We combined the results from the nevus meta-analysis above with results from stage 1 of a recently published meta-analysis of CM[10]. Stage 1 of the CM study consisted of 11 GWAS data sets totaling 12,874 cases and 23,203 controls from Europe, Australia, and the United States; this stage included all six published CM GWAS and five unpublished ones. We do not utilize the results of stage 2 of that study, where a further 3116 CM cases and 3206 controls from three additional data sets were genotyped for the most significantly associated SNP from each region, reaching $P < 10^{-6}$ in stage 1. As a result, certain melanoma association peaks are not genome-wide significant in their own right in the present bivariate analyses. Further details of these studies can be found in the Supplementary Note to Law et al.[10]. The combination of the nevus and melanoma results was performed using the Fisher method. A Manhattan plot for the combined nevus GWASMA plus melanoma GWASMA is shown in Supplementary Fig. 4. For more details of statistical methods, see Supplementary Methods.

## Data availability

All relevant data are available from the authors upon application.

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

## Author contributions

S.C., J.H.B., N.K.H., S.M., A.H., M.K., D.H., J.A.N.B., T.D.S., D.M., G.D.S., T.E.N., D.T.B., V.B., M.F., J.H., and N.G.M. designed the study and obtained funding. G.Z., X.L., M.S., M.I., L.C.J., D.M.E., S.Y., J.B., M.L., P.K., A.V., J.C.T., F.L., J.A.M., and D.G. analyzed the data. M.J.W., E.N., and A.G. contributed to data collection and phenotype definitions. G. W.M., A.K.H., L.B., A.M.I., A.U., P.A.M., A.C.H., E.N., and M.B. contributed to genotyping. D.L.D. and N.G.M. wrote the first draft of the paper. All authors contributed to the final version of the paper.

## Additional information

**Competing interests:** The authors declare no competing interests.

## Melanoma GWAS Consortium

Jeffrey E. Lee[18], Myriam Brossard[19], Eric K. Moses[20], Fengju Song[21], Rajiv Kumar[22], Douglas F. Easton[23], Paul D.P. Pharoah[24], Anthony J. Swerdlow[25], Katerina P. Kypreou[26], Mark Harland[27], Juliette Randerson-Moor[27], Lars A. Akslen[28], Per A. Andresen[29], Marie-Françoise Avril[30], Esther Azizi[31], Giovanna Bianchi Scarrà[32], Kevin M. Brown[33], Tadeusz Dębniak[34], David E. Elder[35], Shenying Fang[18], Eitan Friedman[36], Pilar Galan[37], Paola Ghiorzo[32], Elizabeth M. Gillanders[38], Alisa M. Goldstein[33], Nelleke A. Gruis[39], Johan Hansson[40], Per Helsing[41], Marko Hočevar[42], Veronica Höiom[40], Christian Ingvar[43], Peter A. Kanetsky[44], Wei V. Chen[45], Maria Teresa Landi[33], Julie Lang[46], G. Mark Lathrop[47], Jan Lubiński[34], Rona M. Mackie[48], Graham J. Mann[49], Anders Molven[50], Srdjan Novaković[51], Håkan Olsson[52], Susana Puig[53], Joan Anton Puig-Butille[53], Graham L. Radford-Smith[54], Nienke van der Stoep[55], Remco van Doorn[39], David C. Whiteman[56], Jamie E. Craig[57], Dirk Schadendorf[58], Lisa A. Simms[55], Kathryn P. Burdon[59], Dale R. Nyholt[60], Karen A. Pooley[23], Nicholas Orr[61], Alexander J. Stratigos[26], Anne E. Cust[62,65], Sarah V. Ward[20,66], Hans-Joachim Schulze[63], Alison M. Dunning[24], Florence Demenais[19] & Christopher I. Amos[64]

[18]Department of Surgical Oncology, The University of Texas MD Anderson Cancer Center, Houston, TX, USA. [19]Institut National de la Santé et de la Recherche Médicale (INSERM), UMR-946, Genetic Variation and Human Diseases Unit, Paris, France. [20]Centre for Genetic Origins of Health and Disease, Faculty of Medicine, Dentistry and Health Sciences, The University of Western Australia, Western Australia, Australia. [21]Departments of Epidemiology and Biostatistics, Key Laboratory of Cancer Prevention and Therapy, National Clinical Research Center of Cancer, Tianjin Medical University Cancer Institute and Hospital, Tianjin 300060, P. R. China. [22]Division of Molecular Genetic Epidemiology, German Cancer Research Center, Im Neuenheimer Feld 580, Heidelberg, Germany. [23]Centre for Cancer Genetic Epidemiology, Department of Public Health and Primary Care, University of Cambridge, Cambridge, UK. [24]Centre for Cancer Genetic Epidemiology, Department of Oncology, University of Cambridge, Cambridge, UK. [25]Division of Genetics and Epidemiology, The Institute of Cancer Research, London, UK. [26]Department of Dermatology, University of Athens School of Medicine, Andreas Sygros Hospital, Athens, Greece. [27]Section of Epidemiology and Biostatistics, Leeds Institute of Cancer and Pathology, University of Leeds, Leeds, UK. [28]Centre for Cancer Biomarkers CCBIO, Department of Clinical Medicine, University of Bergen, Bergen, Norway. [29]Department of Pathology, Molecular Pathology, Oslo University Hospital, Rikshospitalet, Oslo, Norway. [30]Assistance Publique–Hôpitaux de Paris, Hôpital Cochin, Service de Dermatologie, Université Paris Descartes, Paris, France. [31]Department of Dermatology, Sheba Medical Center, Tel Hashomer, Sackler Faculty of Medicine, Tel Aviv, Israel. [32]Department of Internal Medicine and Medical Specialities, University of Genoa, Genoa, Italy. [33]Division of Cancer Epidemiology and Genetics, National Cancer Institute, National Institutes of Health, Bethesda, MD, USA. [34]International Hereditary Cancer Center, Pomeranian Medical University, Czechs, Poland. [35]Department of Pathology and Laboratory Medicine, Perelman School of Medicine at the University of Pennsylvania, Philadelphia, PA, USA. [36]Oncogenetics Unit, Sheba Medical Center, Tel Hashomer, Sackler Faculty of Medicine, Tel Aviv University, Tel Aviv, Israel. [37]Université Paris 13, Equipe de Recherche en Epidémiologie Nutritionnelle (EREN), Centre de Recherche en Epidémiologie et Statistiques, Institut National de la Santé et de la Recherche Médicale (INSERM U1153), Institut National de la Recherche Agronomique (INRA U1125), Conservatoire National des Arts et Métiers, Communauté d'Université Sorbonne Paris Cité, F-93017 Bobigny, France. [38]Inherited Disease Research Branch, National Human Genome Research Institute, National Institutes of Health, Baltimore, MD, USA. [39]Department of Dermatology, Leiden University Medical Centre, Leiden, The Netherlands. [40]Department of Oncology-Pathology, Karolinska Institutet, Karolinska University Hospital, Stockholm, Sweden. [41]Department of Dermatology, Oslo University Hospital, Rikshospitalet, Oslo, Norway.

[42]Department of Surgical Oncology, Institute of Oncology Ljubljana, Ljubljana, Slovenia. [43]Department of Surgery, Clinical Sciences, Lund University, Lund, Sweden. [44]Department of Cancer Epidemiology, H. Lee Moffitt Cancer Center and Research Institute, Tampa, FL, USA. [45]Department of Genetics, The University of Texas MD Anderson Cancer Center, Houston, TX, USA. [46]Department of Medical Genetics, University of Glasgow, Glasgow, UK. [47]McGill University and Genome Quebec Innovation Centre, Montreal, Canada. [48]Department of Public Health, University of Glasgow, Glasgow, UK. [49]Centre for Cancer Research, University of Sydney at Westmead, Millennium Institute for Medical Research and Melanoma Institute Australia, Sydney, Australia. [50]Department of Pathology, Haukeland University Hospital, Bergen, Norway. [51]Department of Molecular Diagnostics, Institute of Oncology Ljubljana, Ljubljana, Slovenia. [52]Department of Oncology/Pathology, Clinical Sciences, Lund University, Lund, Sweden. [53]Melanoma Unit, Dermatology Department & Biochemistry and Molecular Genetics Departments, Hospital Clinic, Institut de Investigacó Biomèdica August Pi Suñe, Universitat de Barcelona, Barcelona, Spain. [54]Inflammatory Bowel Diseases, QIMR Berghofer Medical Research Institute, Brisbane, Australia. [55]Department of Clinical Genetics, Leiden University Medical Center, Leiden, The Netherlands. [56]Cancer Control Group, QIMR Berghofer Medical Research Institute, Brisbane, Australia. [57]Department of Ophthalmology, Flinders University, Adelaide, Australia. [58]Department of Dermatology, University Hospital Essen, Essen, Germany. [59]Menzies Institute for Medical Research, University of Tasmania, Hobart, TAS, Australia. [60]Institute of Health and Biomedical Innovation, Queensland University of Technology, Brisbane, QLD, Australia. [61]Breakthrough Breast Cancer Research Centre, The Institute of Cancer Research, London, UK. [62]Cancer Epidemiology and Services Research, Sydney School of Public Health, The University of Sydney, Sydney, Australia. [63]Department of Dermatology, Fachklinik Hornheide, Institute for Tumors of the Skin at the University of Münster, Münster, Germany. [64]Department of Community and Family Medicine, Geisel School of Medicine, Dartmouth College, Hanover, NH, USA. [65]Sydney School of Public Health and the Melanoma Institute Australia, University of Sydney, Sydney, Australia. [66]Department of Epidemiology and Biostatistics, Memorial Sloan Kettering Cancer Center, New York, USA

