## [Peer Review File · Nature Communications]

Reviewers' Comments:

Reviewer #1:

Remarks to the Author:

In the current study, Duffy et al performed a large meta-GWAS analysis of nevus density using 11 independent GWAS datasets of western populations. 6 loci were discovered at genome-wide significance, and three are novel loci (KITLG, DOCK8 and 9q32).

Then, authors performed a joint-GWAS analysis of nevus density and melanoma using a simple Fisher method to combine the P values from the current meta-analysis of nevus density and the ones from a recently published meta-analysis study of melanoma. Authors also investigated pleiotropic effect between the two diseases using GWAS-PW method. 12 loci were found to be melanoma-specific, and 18 loci were found to show pleiotropic effect.

In addition, single variant-based analysis, authors also performed gene-based association test using the combined nevus-melanoma P values and found 5 genes whose evidences survived the correction for multiple testing ($2.3E-6$).

Last, authors performed heritability analysis and found that the overall heritability of nevus count is 58%. They also found that 15-25% of heritability can be explained by 1000 SNPs within 32 regions. They also found that the two diseases show significant sharing of heritability ($r_g = 0.69$).

The study assembled a huge dataset of multiple independent samples and performed a series of analyses. But, the analyses and the results are poorly organized. The manuscript is very hard to follow. And, many important information has not been provided. Here are specific concerns and comments:

1. In order to evaluate the significance of the findings, particularly the novel loci, the information on the impact of population stratification should be provided in the main text. For several novel loci, the evidence was just below the threshold of genome-wide significance. If the population stratification has not been fully controlled, these loci may not be claimed as novel loci reaching genome-wide significance. The results from the meta-analysis of nevus density for the 6 loci reaching genome-wide significance on the page 3 are not consistent with the results presented in the Table 1. In addition, as indicated in the methods, there were significant heterogeneity among the results from the independent samples. It was also stated that both fixed and random effect models were used in the analysis. Which results (in the table 1) were from fixed effect or random effect model? Detailed information of heterogeneity for the 6 loci that were claimed to reach genome-wide significance should be provided.
2. A joint-GWAS analysis of nevus density and melanoma was performed by using a simple Fisher method to combine the P values from the two meta-GWAS analyses. I don't see any value for doing such an analyses, and the results from this analysis was not discussed in the paper. What is the hypothesis that was investigated or tested under such an analysis? how was the heterogeneity issue handled in such analyses?
3. As for the analysis using GWAS-PW method, authors need to clarify why $PPA > 0.5$ for any hypothesis was chosen as the threshold of "interesting". For each of 30 regions that showing the interesting result from the GWAS-PW analysis, how should the statistical significance be evaluated? While $PPA > 0.5$ may suggest that one hypothesis might be more possible than the other hypotheses, what is the overall statistical significance of the analysis? Of these interesting loci, 18 loci were claimed to show significant pleiotropic effect between two diseases. It is not clear what statistical evidence/significance was used to claim these loci. For example, for the pleiotropic loci of TERC, FMN1, SYNE2, the evidences from the nevus density GWAS were far below the threshold of genome-wide significance. Giving that these loci have not been approved as nevus density loci, it is hard to understand how these loci can be claimed as pleiotropic loci between the

two phenotypes. A good volume of write-up/discussion was devoted to the biology implicated by the pleiotropic loci. But, this section may need to be revised significantly after clarifying the statistical issue surrounding the pleiotropic loci.

4. 5 novel genes were discovered by gene-based test using the combined nevus-melanoma meta-analysis P values. Authors only provided the information on which software packages were used for analysis, but did not provide any information about the assumptions/models of these two methods. It is important to provide the results of gene-based test from the separate nevus and melanoma analyses. It is also important to explain how the heterogeneity issue was handled in gene-based test.

5. Authors performed bivariate LD regression analysis. Authors indicated that the nevus density-associated alleles always proportionately showed increased risk for melanoma, but not vice versa. And, as a result, authors claimed that the nevus number is the intermediate phenotype in a causative chain to melanoma originating in all these biologically heterogeneous nevus pathways. First of all, I could not find the relevant data/results in Supplementary Fig 3.7-2 a-c and Fig. 5b. Second, even if the results are true, I am not sure that authors' claim is the only explanation. This result can still be explained as true pleiotropic effect loci where the genes play important role in the development of two disease phenotypes, but the two disease phenotypes are not clinical related (meaning that the onset of one disease will lead to the development of the second disease).

Reviewer #2:

Remarks to the Author:

This is a nice meta-analysis of several nevus count GWAS studies with a detailed description of methodology in the supplement.

Additionally, the authors include an analysis with melanoma GWAS studies to identify SNPs associated with nevus and/or melanoma.

By combining these multiple studies, these authors are able to identify additional SNPs of significance.

The authors do a nice job of making their findings concise and highlighting interesting biologic features.

The authors should mention that their study is limited to Caucasian populations in the main text in addition to the supplement.

Thank for the opportunity to revise this MS. We realized that most of the points raised by Reviewer 1 were a consequence of the compressed format we used in our original submission as a letter to Nature Genetics and that the points the reviewer raised were nearly all covered in the extensive OSI we submitted with the main text. Consequently, taking advantage of the more relaxed requirements of Nature Communications, we have moved quite a bit of material from OSI to the main text to satisfy the points raised by Reviewer 1.

Overall, we have reorganised the article into the Nature Communications format, with a separate Results, Discussion and Methods. We have expanded the introduction, created an abbreviated description of the methods, and provided an expanded discussion of the loci we have detected.

We have also taken the opportunity to cite a skin pigmentation GWAS meta-analysis of ours that is to appear in Nat Comm, where two of our novel nevus loci appear.

We will address the comments of the reviewers point by point.

Reviewer 1

But, the analyses and the results are poorly organized. The manuscript is very hard to follow. And, many important information has not been provided. Here are specific concerns and comments:

1. In order to evaluate the significance of the findings, particularly the novel loci, the information on the impact of population stratification should be provided in the main text. For several novel loci, the evidence was just below the threshold of genome-wide significance. If the population stratification has not been fully controlled, these loci may not be claimed as novel loci reaching genome-wide significance.

We are afraid that the heterogeneity test results only appear in Section 5 of the OSI, where they accompany the forest plots. We have now also included these together in a new table in the OSI S3.2-1. In the main text, we mention that heterogeneity in association with nevus count was only substantial for *IRF4* and *MTAP* – in both cases, the overall main effect is extremely significant, so we are not concerned that the previously published identification of these loci as melanoma and nevus QTLs is incorrect. We highlighted these genes because the between-population heterogeneity is an interesting population genetic characteristic rather than a statistical nuisance. In the case of *IRF4*, there are very large allele frequency clines across Europe and within the British Isles, a feature that has occasionally caused it to be removed from GWAS datasets for failing quality control tests. In addition, in our earlier paper, we demonstrated extreme age x genotype interaction, with reversal of sign of the regression coefficient in older versus younger individuals. This is replicated in the present paper in the OSI Section S4.2. In the case of *MTAP*, the level of significance is more due to the statistical power (given the size of the main effect).

The results from the meta-analysis of nevus density for the 6 loci reaching genome-wide significance on the page 3 are not consistent with the results presented in the Table 1.

It is stated in the legend to Table 1 that:

“The SNP with the smallest combined P value under each peak has been selected, but the table rows are ordered by strength of association to nevus count.” In several cases, this is not the most significantly associated SNP for nevus count alone.

We are confident that combining the evidence from melanoma and from nevus count gives us greater resolution to pick out the best candidate for a causal or at least best tagging SNP in each region.

In addition, as indicated in the methods, there were significant heterogeneity among the results from the independent samples. It was also stated that both fixed and random effect models were used in the analysis. Which results (in the table 1) were from fixed effect or random effect model?...Detailed information of heterogeneity for the 6 loci that were claimed to reach genome-wide significance should be provided.

As noted above, these appear in the OSI. Given space constraints, we did not include them in the main text.

2. A joint-GWAS analysis of nevus density and melanoma was performed by using a simple Fisher method to combine the P values from the two meta-GWAS analyses. I don't see any value for doing such an analyses, and the results from this analysis was not discussed in the paper. What is the hypothesis that was investigated or tested under such an analysis? how was the heterogeneity issue handled in such analyses?

Paragraph 2 of the main text explicitly states “combined these results with those from a recently published meta-analysis of melanoma GWAS to increase power to detect pleiotropic genes.” Given that we state in the introduction that nevus count is the greatest measurable phenotypic risk factor for melanoma in European populations, and that nevus count is highly heritable, we thought it unnecessary to explicate why we thought a combined analysis is not only appropriate, but also efficacious in detecting novel loci. We understand that in preliminary analyses from other groups, the novel loci we have highlighted will probably all replicate. We have expanded the description in paragraph 2 to read:

“Total nevus count is highly heritable (60-90% in twins), but only a small proportion of this genetic variance is explained by loci identified so far. The known nevus count loci all have pleiotropic effects on CM risk, which implies both that nevus count loci are medically important, and that a genetic analysis combining nevi and CM phenotypes will have increased statistical power. Here we present a new large nevus genome-wide association meta-analysis, and combine these results with those of a previously published similar melanoma meta-analysis.”

3. As for the analysis using GWAS-PW method, authors need to clarify why $PPA > 0.5$ for any hypothesis was chosen as the threshold of “interesting”. For each of 30 regions that showing the interesting result from the GWAS-PW analysis, how should the statistical significance be evaluated? While $PPA > 0.5$ may suggest that one hypothesis might be more possible than the other hypotheses, what is the overall statistical significance of the analysis?

Again space considerations did not allow us to fully discuss Bayesian philosophy in the main text. The “interesting” threshold of 0.5 is actually that used by Giambartolomei and coworkers (see Ref 38 in the OSI) to define “reliable evidence” of a shared causal variant in the Bayesian approach the GWAS-PW program is an extension of.

...the method provides reliable evidence to detect a shared causal variant behind the GWAS and another trait (median posterior probability of any hypothesis >50%)...

Giambartolomei C et al. A Bayesian Framework for Multiple Trait Colocalization from Summary Association Statistics. bioRxiv preprint first posted online Jun. 26, 2017; doi: <http://dx.doi.org/10.1101/155481>.

Giambartolomei C et al. Bayesian Test for Colocalisation between Pairs of Genetic Association Studies Using Summary Statistics. PLOS Genetics (2014) <https://doi.org/10.1371/journal.pgen.1004383>

The threshold also captured those regions that we had already highlighted looking at the frequentist meta-analysis and using a combined P-value of $5e-8$. The strength of the Bayesian approach is that the posterior probability does have an intuitive interpretation, and allow the reader to carry out their own decision analysis regarding whether such peaks are worthy of follow-up – a pragmatic measure of significance.

Of these interesting loci, 18 loci were claimed to show significant pleiotropic effect between two diseases. It is not clear what statistical evidence/significance was used to claim these loci. For example, for the pleiotropic loci of TERC, FNMI, SYNE2, the evidences from the nevus density GWAS were far below the threshold of genome-wide significance. Giving that these loci have not been approved as nevus density loci, it is hard to understand how these loci can be claimed as pleiotropic loci between the two phenotypes

In the case of *TERC*, *FNMI* and *SYNE2*, the GWAS-PW posterior probabilities of pleiotropy were 97%, 98% and 86% respectively (Figure 3 and Supplementary Table S3.6-1). These probabilities are already allowing for the presence of multiple trait loci.

In a frequentist framework, a P-value for association to a second trait conditional on the fact that one trait has already reached genome-wide significance would require only a multiplicity adjustment for number of initial hits – so obviously the association signal for each of the two traits does not have to reach an individual genome-wide significance level.

Under the null hypothesis that a SNP is not associated to either trait, the Fisher P-value based test has the correct size. This suggests to me (DLD) that the correct threshold adjustment for each trait test **conditional on the combined P-value exceeding the genome-wide corrected threshold** might be $2 * (\text{number of genome-wide significant loci})$, ie $0.05 / (2 * 18) = 1.4e-3$. The maximum P-value (across traits) for *TERC*, *FNMI* and *SYNE2* are $1.6e-5$, $8e-6$ and $1.3e-4$.

A good volume of write-up/discussion was devoted to the biology implicated by the pleiotropic loci. But, this section may need to be revised significantly after clarifying the statistical issue surrounding the pleiotropic loci.

I hope that this is addressed by the above answers. We also now point out that two of our novel loci are also pleiotropic for skin colour in our very-soon-to-appear Nature Communications meta-analysis of skin colour:

Visconti, A., Duffy, D.L. et al. Genome-wide association study in 176,678 Europeans reveals genetic loci for tanning response to sun exposure. Nat Commun (2018)

4. 5 novel genes were discovered by gene-based test using the combined nevus-melanoma meta-analysis P values. Authors only provided the information on which software packages were used for analysis, but did not provide any information about the assumptions/models of these two methods. It is important to provide the results of gene-based test from the separate nevus and melanoma analyses. It is also important to explain how the heterogeneity issue was handled in gene-based test.

We have performed gene-based testing using standard packages that combine the association statistics from our study with the known linkage disequilibrium correlation between SNPs in our European populations. The programs are referenced appropriately in the OSI.

5. Authors performed bivariate LD regression analysis.

The authors did carry out a bivariate LD regression, but this gave very disappointing results – markedly underestimating the “chip” heritability of nevus count. This is a very minor part of the analyses.

Authors indicated that the nevus density-associated alleles always proportionately showed increased risk for melanoma, but not vice versa. And, as a result, authors claimed that the nevus number is the intermediate phenotype in a causative chain to melanoma originating in all these biologically heterogeneous nevus pathways.

First of all, I could not find the relevant data/results in Supplementary Fig 3.7-2 a-c and Fig. 5b. Second, even if the results are true, I am not sure that authors' claim is the only explanation. This result can still be explained as true pleiotropic effect loci where the genes play important role in the development of two disease phenotypes, but the two disease phenotypes are not clinically related (meaning that the onset of one disease will lead to the development of the second disease).

I am guessing the reviewer is referring to what we termed a “Genetic Factor model” as opposed to “Direction of Causation model” [[10.1002/gepi.1370110606](https://doi.org/10.1002/gepi.1370110606)], that is, that an unmeasured intermediate trait under genetic control is a proximal cause of both measured phenotypes. The observed trait most strongly correlated with the latent trait acts as an instrumental variable, and appears to cause the second trait. Yes, it is not completely foolproof – a fact which we now mention in the Discussion. We have included a new figure Fig 3.7-2 that summarizes the correlation between nevus and CM signals for the 21 top nevus SNPs.

Reviewer #2 (Remarks to the Author):

The authors should mention that their study is limited to Caucasian populations in the main text in addition to the supplement.

Yes, we have now highlighted this in the main text.

We hope that these changes are satisfactory,

SECOND RESPONSE TO REVIEWERS (NCOMMS-17-19707-T) 2018-Jun-04

We thank the reviewers for their forbearance.

Reviewer #1 (Remarks to the Author):

1. Good to see that authors have done a deeper evaluation of heterogeneity issue.

Our view is that there is some misunderstanding about the role of random effects models for between-population heterogeneity in the setting of genetic association. There seems to be a certain fixation on the concept of genetic heterogeneity as a statistical nuisance. In the setting of a drug trial, say, it makes sense to get a correct estimate of the average effect size across different settings. But in a population genetic context, if there is statistically significant evidence of differences between populations in strength of association, then we have detected a real biological phenomenon, providing we can exclude methodological artefact as the cause e.g. Wen and Stephens [*Ann Appl Stat*, 8: 176–203, 2014] “it is of considerable interest to identify genetic variants that show association in any subgroup, or in other words to reject the ‘global’ null hypothesis of *no association within any subgroup* [emphasis in original].” And see Lebrech et al [*Stat Appl Genet Mol Bio*; 2010], Han and Eskin [*Am J Hum Genet* 88:586-598, 2011], Neupane et al [*Eur J Hum Genet* 20: 1174–1181, 2012]. To this way of thinking, the detection of marked heterogeneity for *IRF4* gives strong support for association of rs12203592 genotype with nevus count ***in some but not all populations***. The fact that the mean effect size is zero is a consequence of the choice of populations to study. Given that many of the nevus-associated loci are involved in pigmentation, it is not surprising these exhibit between-population differences in association – both allele frequency and linkage disequilibrium pattern differences can lead to this, quite aside from gene by environment interaction.

Han and Eskin [*Am J Hum Genet* 88:586-598, 2011] have demonstrated that the test of significance of the overall mean association in a standard random effects model is quite conservative. This result is confirmed in their simulations and also those presented by Neupane et al [*Eur J Hum Genet* 20: 1174–1181, 2012]. This is the underlying reason that the current “standard” approach has been to prefer a fixed effects analysis over random effects unless there is a significant evidence of heterogeneity. In this two-stage approach, since the heterogeneity test is being performed across multiple loci, some adjustment for multiple testing should be made but to be conservative we have simply applied the uncorrected nominal P value of 0.05

Han and Eskin [2011] suggested an alternative random effects test, as implemented in the METASOFT package (<http://genetics.cs.ucla.edu/meta/>), which tests the composite hypothesis that the mean effect and heterogeneity are zero. Han and Eskin, and also Neupane et al (*loc cit*) show that this new test has a correct type 1 error rate and is more powerful in the presence of heterogeneity than the standard random effects approach. We compare the Han and Eskin P-values to those from other approaches in the table below (which has now been included in OSI as S3.2.2). In most cases, the Han and Eskin test agrees with the

conventional fixed effects analysis P, except for rs12203592 in *IRF4* where it is much more significant. In the specific case (raised by Reviewer 1) of rs600951 in *DOCK8*, the Han and Eskin random effects model is $P=2 \times 10^{-8}$.

One suggestion of Lebec et al [2010] was to combine the association Wald Z^2 scores for each study as chi-squares, with the degrees of freedom being the number of studies. This was characterized by Neupane et al [2012] as a fixed effects test allowing for heterogeneity. It is very similar to the Fisher method in spirit and in the results (see 5th v. 6th columns of table below, from which it is evident that this is a low power test).

Consequent on the above, we have now changed Table 2 in the main text to give for nevus count the fixed effects meta-analysis P values where the homogeneity P is $>.05$ and the Han-Eskin random effects P value when homogeneity $P < .05$. In the latter case we also present the fixed effects P value for the SNP nearest to the peak where the homogeneity chi-square is not significant, in order to confirm that the association is genuine and not inflated by stratification. The consequences of this slight change in our approach (in response to the concerns of Reviewer 1) are minimal. The text has been correspondingly amended (Results, first paragraph, Methods, and also OSI).

(new) Table S3.2.2. Comparison of different meta-analysis P-values for top SNPs associated with either nevus count. Note that P-values for the Hans & Eskin random effects model are in general more significant for our top loci, notably *IRF4*, *MTAP* and *DOCK8* where between sample heterogeneity is highest.

SNP	Position (B37)	N studies	Heterogeneity I ² (P-value)	Tests of association by different methods				
				Fisher P	Lebrec FE P	H&E RE P	RE P	FE P
rs4670813 (CYP1B1)	2:38317710	11	0% (0.585)	3.94E-04	4.42E-04	1.09E-06	5.70E-07	5.70E-07
rs55875066 (HDAC4)	2:240076002	11	25% (0.203)	1.46E-04	8.35E-05	1.01E-06	4.30E-05	7.58E-07
rs12696304 (TERC)	3:169481271	11	0% (0.719)	3.43E-03	3.67E-03	1.07E-05	5.73E-06	5.73E-06
rs251464 (PPARGC1B)	5:149196234	11	40% (0.083)	1.55E-05	1.60E-05	7.08E-07	6.53E-04	4.72E-07
rs12203592 (IRF4)	6:396321	10	97% (3.35E-51)	1.24E-64	1.52E-66	4.21E-67	3.12E-01	2.54E-18
rs600951 (DOCK8)	9:224742	11	68% (5.86E-04)	3.66E-08	8.73E-08	1.95E-08	9.86E-03	1.12E-06
rs869329 (MTAP)	9:21804693	11	75% (1.72E-05)	7.88E-35	5.80E-35	2.12E-37	1.84E-08	1.11E-34
rs1484375 (9q31.1)	9:109067561	10	0% (0.713)	2.47E-02	2.23E-02	2.36E-04	1.35E-04	1.35E-04
rs10816595 (9q31.2)	9:110709735	11	34% (0.126)	1.14E-06	1.50E-06	1.94E-08	6.83E-06	1.08E-08
rs45575338 (FAM208B)	10:5784151	11	0% (0.805)	6.87E-03	7.47E-03	1.89E-05	1.02E-05	1.02E-05
rs1640875 (GPRC5A)	12:13069524	11	55% (0.015)	6.30E-05	3.27E-05	5.72E-06	8.21E-03	2.08E-05
rs7313352 (KITLG)	12:88949124	11	0% (0.476)	7.67E-06	1.18E-05	1.56E-08	8.40E-09	8.40E-09
rs2357176 (SYNE2)	14:64409313	11	12% (0.331)	1.10E-02	8.45E-03	3.43E-04	6.55E-04	1.95E-04
rs117648907 (FMN1)	15:33277710	4	0% 0.436	1.05E-04	1.24E-04	9.50E-06	6.52E-06	6.52E-06
rs34466956 (NFIC)	19:3353622	10	42% (0.077)	1.03E-03	1.16E-03	3.85E-04	4.06E-02	2.22E-04
rs132985 (PLA2G6)	22:38563471	11	69% (4.23E-04)	1.96E-16	9.39E-17	3.06E-18	6.53E-05	7.45E-17

Note: FE – Conventional fixed effects; Lebrec FE – Lebrec et al (2010) fixed effects test incorporating heterogeneity; HE - Han and Eskin random effects model (2011); RE - random effects. Bold type indicates genome-wide significance

Authors comments that the significant heterogeneity was observed for IRF4 and MTAP, but this will not cause any concerns on the overall evidences for these two loci due to their strong genetic effect. While agreeing with the authors on the MTAP locus (still reaching genome-wide significance under a random effect model), I am not sure about the IRF4 locus. Although the evidence under the fixed effect model is highly significant, there is huge evidence for heterogeneity, and there is no evidence for association under the random effects model. Authors claimed that the huge heterogeneity is due to the age x genotype interaction. In this case, authors should do stratified analysis by age group to demonstrate whether the association can be replicated.

We draw the reviewer's attention to Table S4.2-1 in Section S4.2 of the Supplementary Information, where the association P-value for nevus count and rs12203592 is $P=10^{-31}$ just using a single sample of Australian adolescents; this obviates any effects of age and insolation. The same table demonstrates that the regression coefficient for the parents of *the same twins* was opposite in sign in a bivariate family-based analysis. We have not concentrated on this in the main text given this is a known locus and that we have told this story fully in a previous paper [Duffy et al, *Am J Hum Genet* 87: 6–16, 2010] that focuses on just this issue. We also, in Table S4.3-1 present results of a melanoma-nevus bivariate mixed-model analysis with the melanoma pedigrees from the Queensland Study of Melanoma: Environmental and Genetic Associations (Q-MEGA) study [Baxter et al, *Twin Res Hum Genet* 11:183-196, 2008]. This shows that in adults, the rs12203592*T allele is associated both with increased risk of melanoma and decreased nevus count. In the adolescents, the rs12203592*T allele was associated with increased nevus count. It is the strength of the association within just this one population that leads to the heterogeneity test in the meta-analysis being so significant.

Actually, significant heterogeneity was also observed in the DOCK8 and PLA2G6. In particular, the evidence for the DOCK8 does not reach genome-wide significance under the random effects model. And, for the loci showing significant heterogeneity, the results from the random effects model (instead of the fixed effects) should be provided in the Table 2 (NevusP), and the statistical evidence from the random effects model should be used for the combined analysis of Nevus and CM GWAS.

The PLA2G6 SNP rs132985 reaches $P=1.6 \times 10^{-8}$ and $P=2.2 \times 10^{-8}$ in the Harvard and TwinsUK studies respectively for association to nevus count. Evidence of an effect on melanoma risk is equally strong. Our interpretation of these results is that PLA2G6 is a nevus and melanoma locus but that effects vary significantly across populations. Regarding the case of DOCK8, see discussion above.

2. For the combined analysis of the Nevus and CM GWAS results, was the heterogeneity tested between Nevus and CM results? If there is significant heterogeneity, I am not sure whether Fisher method should be used for the combined analysis.

It is not possible to formally test for heterogeneity with only two studies. However, we do explore this issue qualitatively in the Miami plot shown as Figure 1 in the main text, and also by regressing betas for nevi against the corresponding betas for melanoma in Supp Figure S3.6-2 which demonstrate pleiotropy for most loci except those that we specifically note; these are further explored in Miami plots of each locus in Supp Figures Section 5.1.

This issue is particularly important for the novel loci HDAC4, GPRC5A, FAM208B, NFIC and SYNE2 where only the evidence from the combined analysis (Fisher P) reached genome-wide significance, and some of these evidences (Fisher P) were just above the threshold of genome-wide significance. Without a proper control on the heterogeneity issue, the evidence from the Fisher analysis suffers the concern of inflation.

This is why we also present the Bayesian results from our analyses using GWAS-PW (page 3-5 and Figure 3 in main text, plus Supp Section 3.7) – to show that both methodologies converge to the same conclusions. The advantage of the Bayesian method is that it uses the empirical distribution of P-values across the different LD regions to give a corrected posterior P-value.

3. Authors failed to address my original comment on population stratification. The lambda value needs to be provided for the whole meta-GWAS analysis (in addition to the information for each GWAS dataset and analysis).

Sorry about that. The overall lambda for the nevus meta-analysis is 1.40, and the standardized $\lambda_{1000} = 1.008$ – in keeping with the appearance of Figure S3.2-2 and consistent with the genomic inflation factor expected under polygenic inheritance with the observed heritability and sample size [Yang et al, *Eur J Hum Genet* 9:807-12, 2011]. We have added this to the main text. For nevus plus melanoma, $\lambda=1.36$ and $\lambda_{1000} = 1.004$ (added to OSI).

Taking together, I am not convinced that 8 novel loci were discovered in this study.

The combination of two P values by the methods in the manner we have used has the correct type 1 error rate under the null hypothesis. In passing, we have reanalysed using the Stouffer method rather than the Fisher method because it is known to be slightly more powerful (see eg the review by Zaykin, *J Evol Biol* 24:1836-41, 2011). Choosing a more conservative threshold for genome wide significance would decrease the number of loci we declare, but simultaneously will reduce the statistical power. Given that we present the evidence supporting each locus, the reader will be well able to assess the strength of evidence at any given locus which may be of interest to them.

Reviewer #2 (Remarks to the Author):

Page 4: ref to DOCK8 deficiency:

Have added refs to:

DOCK8 deficiency: Insights into pathophysiology, clinical features and management. Biggs CM, Keles S, Chatila TA. *Clin Immunol*. 2017 Aug;181:75-82.

Cancer Genome Atlas Network et al: Genomic Classification of Cutaneous Melanoma Cell 2015
61:1681-96

Page 4 foot: "...several different pathways implicated which are all expressed in melanocytes":
Unclear meaning; do you mean there are lines of evidence that indicate these genes are expressed in melanocytes?

For those loci that unequivocally lie within a gene, those genes are all expressed in melanocytes and implicate several different pathways. We have clarified this in the text.

Page 6 foot: GTEx eQTL

Expand these abbreviations and perhaps a bit on their biological significance: Done

Table 2: cutaneous melanoma (CM)

BP38 change to hg38 position: done

Figure 3: add y-axis labels – PPAa etc: done

Figure 5: *correct UCSC*; done

THIRD RESPONSE TO REVIEWERS (NCOMMS-17-19707-T) 2018-Jun-24

The editor has asked us “...to consider your response to referee 1's third point (please see below) and consider the minor textual edit suggested by referee 2 in the form of a revised manuscript...”

Reviewer #1 (Remarks to the Author):

3. *As for the population stratification vs polygenetic inheritance, authors might want to do a formal evaluation using LD score regression based method (Nature Genetics 2015 47: 291 - 295).*

The reviewer has momentarily forgotten that we discussed this in our Response Revision 1. This is addressed in Sections S2.3.2 and S3.4.3 of the OSI (which is unchanged from December 2017):

“S2.3.2 *Bivariate heritability analysis of melanoma and nevus count*

We also performed analyses examining the overall architecture of the relationship between nevus count and melanoma risk. First, a bivariate REML analysis using the GCTA package of a melanoma Queensland case-control family-based sample that overlaps one study used by Law and coworkers; there were 5,210 individuals from 3,520 melanoma case and control families, including 1,137 melanoma cases, and nevus assessment was a 4-point self-rating questionnaire item. We compared these results to those of a bivariate LD score regression analysis of summary results from the entire nevus and melanoma meta-analyses (<https://github.com/bulik/ldsc>).

S3.4.3 *Bivariate heritability analysis of melanoma and nevus count*

...We also carried out bivariate LD score regression analysis of the nevus and melanoma meta-analyses...For melanoma, [a Queensland-]family-based GCTA and [the] total dataset LD score analyses gave somewhat different point heritability estimates of 0.59 (0.22) and 0.16 (0.03) respectively. The latter may be closer to the SNP heritability h^2_s and is consistent with the h^2_s estimate of 19.2% due to significant SNPs in the Law et al meta-analysis, while the former estimate is consistent with the value for melanoma age of onset of 0.45 (0.05) obtained using a more elaborate method on an overlapping data-set.

Both analyses badly underestimated the h^2 of nevus count at 0.07 (0.04) and 0.03 (0.01) respectively. This is probably because of the coarse nevus measure in the [melanoma] family study and the genetic architecture for the LD score analysis (the original authors note that the method does not seem to "work" for some phenotypes, including perhaps melanoma, as above). In both analyses, however, the r_g is high at 0.68 (0.40) and 0.66 (0.14), albeit with large SE for the pedigree estimate. We interpret this as reflecting the strong genetic causation of nevus number along with a direct phenotypic causal pathway to melanoma, that is, the genes we have found for nevus number act via different pathways but will all affect melanoma risk...”

One assumption of the LDSC methods is that the trait is markedly polygenic, so contributions come from all regions, and weighting of regions is thus uniform. It is possible that the architecture of pigmentation related traits differs from this because of selection – we know there is strong selection on several of our key loci, notably *IRF4* and *MC1R*. We also note also that the LD Score website (<https://github.com/bulik/ldsc/wiki>) says “[s]ummary statistics from linear mixed models cannot be used to estimate SNP-heritability”. Given that the Australian and Twins UK results are linear mixed model analyses because both are twin-family based samples, this might explain the low estimates of the heritability from LD score regression. If one applies LD score regression to individual substudies, one obtains for example:

Harvard 0.0462 (0.0145), ALSPAC -0.0263 (0.1315), BTNS 0.2611 (0.1894). The LDSC manual mentions that “As a rule of thumb, LD Score regression tends to yield very noisy results when applied to datasets with fewer than ~5k samples, even for univariate h^2 estimation.”

Reviewer #2 (Remarks to the Author):

Additional optional edit:

"Their association with CM can be direct, in that a proportion of melanomas arise within a pre-existing nevus (due to a "second hit" mutation),"

should be changed in my opinion to

"Their association with CM can be direct, in that a proportion of melanomas arise within a pre-existing nevus (due to additional mutations),"

The "second-hit" typically refers to loss of the second copy of a tumor suppressor, which does not apply here. Also we believe more than one additional oncogenic mutation is required for transformation of a nevus to melanoma.

This is why this was in quotation marks, but I think the reviewer's rewording sounds better, and we have made this change.

I hope this clears up any remaining concerns.

Yours,

David Duffy.